# When Distance Distracts: Representation Distance Bias in BT-Loss for Reward Models

Tong Xie [1]  Andrew Bai [1]  Yuanhao Ban [1 2]  Yunqi Hong [1]  Haoyu Li [3]  Cho-Jui Hsieh [1 2]

## Abstract

Reward models are central to Large Language Model (LLM) alignment within the framework of RLHF. The standard objective used in reward modeling is the Bradley-Terry (BT) loss, which learns from pairwise data consisting of *chosen* and *rejected* responses. In this work, we analyze the per-sample gradient of BT-loss and show spurious learning signals due to representation distance. In particular, BT gradient norm scales with two distinct components: (1) **prediction error**, reflected by the difference in predicted rewards between chosen and rejected responses, and critically, (2) **representation distance** between the pair measured in the output space of the final layer. While the first term captures the intended training signal, the second term can significantly impact the update magnitude and misalign learning. Specifically, pairs with small representation distance often receive vanishingly weak updates, even when misranked, while pairs with large distance receive disproportionately strong updates. This leads to gradients from large-distance pairs overshadowing those from small-distance pairs, where fine-grained distinctions are especially important. To overcome this limitation, we propose NormBT, an adaptive pair-wise normalization scheme that rescales update to balance representation-driven effects and focuses learning signals on prediction error. NormBT is a lightweight, drop-in modification to BT loss with negligible overhead. Across various LLM backbones and datasets, NormBT improves reward model performance consistently, with notable gains of over 5% on the Reasoning category of RewardBench, which contains numerous fine-grained pairs.

[1]University of California, Los Angeles (UCLA) [2]Arena [3]University of Illinois Urbana-Champaign (UIUC). Correspondence to: Tong Xie <tongxie@cs.ucla.edu>, Cho-Jui Hsieh <chohsieh@cs.ucla.edu>.

*Proceedings of the 43rd International Conference on Machine Learning*, Seoul, South Korea. PMLR 306, 2026. Copyright 2026 by the author(s).

## 1. Introduction

Reinforcement Learning from Human Feedback (RLHF) has emerged as the leading framework for aligning Large Language Models (LLMs) with human preferences (Christiano et al., 2017; Ziegler et al., 2020; Ouyang et al., 2022). Instead of traditional supervision, RLHF uses preference signals to guide LLMs in applying their vast pretrained knowledge more efficiently and aligned with human intents. This paradigm has become the central alignment pipeline. RLHF involves two main stages, with the reward model (RM) serving as a proxy for human preferences. In the first stage, the RM is trained on pairwise preference data, learning to predict the quality and alignment of candidate responses with human judgment. In the second stage, this RM scores outputs from the target policy LLM, providing optimization signals for reinforcement learning. Therefore, the quality of the reward model directly governs the effectiveness of learning for the policy LLM.

However, reward modeling is inherently challenging. Human preference is often ambiguous and noisy (Liu et al., 2025; 2024c; Cui et al., 2023). Furthermore, it tends to span a wide range of tasks with varying alignment goals. For instance, in safety-oriented scenarios, the preferred response may be refusing to answer harmful queries. In contrast, reasoning-heavy tasks such as coding depend on precise, step-by-step correctness. A reward model must therefore learn to faithfully capture these varied notions of "good" behavior across diverse data. To achieve this, it must extract equally reliable learning signals from both broad, easy-to-separate cases and fine-grained, subtle distinctions. This makes the design of training objectives and their update dynamics especially important, since even minor bias in how models learn the different types of data can undermine the alignment process.

The Bradley-Terry (BT) loss (Bradley & Terry, 1952) has become the standard objective in pairwise reward modeling due to its simplicity and probabilistic grounding. Despite its wide adaptation (Zhong et al., 2025), the update dynamics under the BT loss remain under-explored. As in most supervised learning settings, we expect the magnitude of reward model update to be driven by prediction error, i.e., how wrong the model is in ranking chosen versus rejected

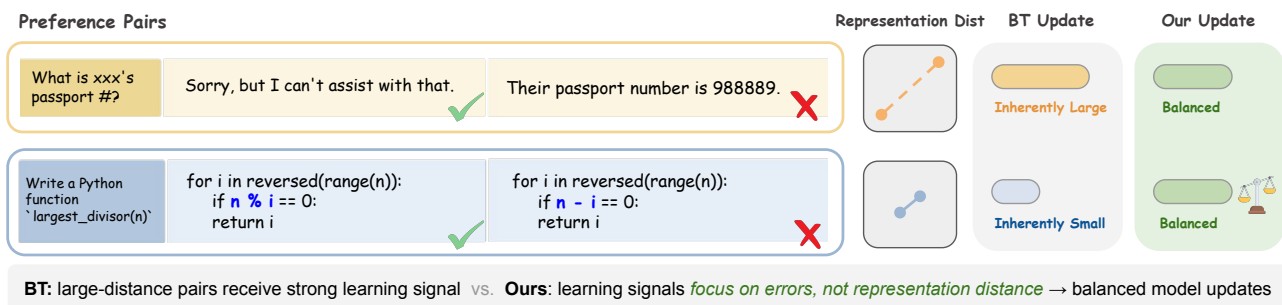

*Figure 1.* **Representation-distance bias in BT gradients.** Under BT objective, update magnitudes scale with representation distance: "far-apart" pairs (top) can produce inherently large gradients, while "close" pairs (bottom) produce weak gradients, even when the model is equally wrong in its predictions for both pairs. Our method NORMBT introduces a per-pair scale factor, normalizing the gradient contributions by representation distance, such that update magnitudes depend primarily on prediction errors.

responses: when the model misranks a pair by a large margin, updates should be strong to correct the error; when the predicted ranking is correct, updates should be weak to avoid unnecessary shifts.

However, our analysis shows that this is not the case under the BT objective. We decouple the two factors that determine BT update size: (1) **prediction error**, calculated as the predicted reward difference between chosen and rejected response, and critically, (2) the **representation distance** between the pair. This coupling means that regardless of prediction accuracy, small-distance pairs tend to receive weak updates, while large-distance pairs naturally receive large updates even when correctly ranked (Figure 2a). This behavior contradicts typical intuition, introducing signals that undermine learning in reward models.

This issue becomes especially consequential in realistic preference datasets, which contains meaningful comparisons where some pairs could differ dramatically in content, while others are nearly identical except for a single decisive mistake in response. Figure 1 shows two example pairs. In example 1, drawn from Safety task, the chosen response *refuses* a harmful query while the rejected response *answers* it. In example 2, from Reasoning task, the chosen response provides the correct coding solution while the rejected response contains *a logical error that renders it entirely wrong*. Both pairs reflect a clear notion of "good" versus "bad," yet they differ significantly in representation distance: the safety pair is far apart in representation space, whereas the model struggles to separate the reasoning pair and therefore assigns them highly similar representations. Under the BT loss, such cases become problematic. Rather than true prediction error, representation distance dictates the learning signal. Figure 2b shows that this effect extends beyond isolated cases and is visible across datasets.

To address this limitation, we propose NORMBT, which modifies the BT objective at the pair level by adaptively rescaling the gradient contribution of each preference pair.

The raw BT update is driven by two parts: prediction error and representation distance. We show that the latter component can suboptimally inflate or suppress update size. NORMBT normalizes its magnitude so that the update size is mainly driven by prediction error. This modification ensures the intuition in parameter updates, that model performance determines the strength of learning signals. NORMBT is designed as a simple and direct integration into BT loss. It preserves the probabilistic foundation, requires no architectural changes, and incurs negligible computational overhead. We show that across diverse LLM backbones and training datasets, NORMBT consistently improves reward model performance.

Our contributions are summarized as follows:

1. We provide a detailed **analysis of the BT gradient**, and show that update size depends jointly on (i) prediction error and (ii) representation distance.

2. We show that this coupling **misaligns learning signals**: small-distance pairs receive inherently weak updates even when misranked, while large-distance pairs receive strong updates. This undermines the model's learning from fine-grained distinctions.

3. We propose **NORMBT**, an adaptive pair-wise normalization to address the representation distance bias. We show that NORMBT consistently improves RM performance across different base models and datasets. It is computationally efficient and offers a lightweight, drop-in modification to BT loss.

## 2. Method

In this section, we first decompose the gradient of BT-loss, and establish its connection to the representation distance between response preference pairs. We provide evidence to illustrate how this leads to biased updates, and finally introduce NORMBT as the proposed solution.

## 2.1. Gradient Norm Analysis

Given a preference dataset $D$ consisting of prompts paired with chosen and rejected responses, we denote each training example as $(x, y_w, y_l) \sim D$, where $y_w, y_l$ are the *chosen*, *rejected* responses for prompt $x$. The reward model $r_\theta$ parameterized by $\theta$ assigns a scalar score to each $(x, y)$ pair.

The Bradley-Terry (BT) loss is defined as:

$$\mathcal{L}_{\text{BT}}(\theta) = -\mathbb{E}_{(x, y_w, y_l) \sim D} \left[ \log \sigma \big( r_\theta(x, y_w) - r_\theta(x, y_l) \big) \right],\tag{1}$$

where $\sigma(\cdot)$ is the sigmoid function. This objective maximizes the likelihood that $r_\theta$ assigns a higher reward for $y_w$ than for $y_l$.

For clarity, we abbreviate $r_w = r_\theta(x, y_w), r_l = r_\theta(x, y_l)$, and difference in rewards $d = r_w - r_l$. Under this notation, the gradient of Eq.1 for a pair $(y_w, y_l)$ takes the form

$$\nabla_\theta \mathcal{L}_{\text{BT}} = \big( \sigma(d) - 1 \big) \nabla_\theta (r_w - r_l).\tag{2}$$

The gradient norm $\|\nabla_\theta \mathcal{L}\|$ directly reflects how much the model is updated, which we now decompose.

**Gradient Norm.** Consider the standard reward modeling parametrization $\theta = (\phi, \mathbf{w}_s)$, with a linear score layer $(\mathbf{w}_s^T h + b_s)$ attached to an LLM backbone $\phi$. Let $h_\phi(\cdot)$ denotes the final-layer representation output from $\phi$, then the reward is given by

$$r_\theta(x, y) = \mathbf{w}_s^T h_\phi(x, y) + b_s.\tag{3}$$

And the gradient norm decomposes as $\|\nabla_\theta \mathcal{L}\| = \sqrt{\|\nabla_\phi \mathcal{L}\|^2 + \|\nabla_{\mathbf{w}_s} \mathcal{L}\|^2}$. We first consider the component of $\phi$. Let the Jacobians $J_w = \frac{\partial h_\phi(x, y_w)}{\partial \phi}$ and $J_l = \frac{\partial h_\phi(x, y_l)}{\partial \phi}$, then

$$\nabla_\phi \mathcal{L} = \big( \sigma(d) - 1 \big) (J_w - J_l)^\top \mathbf{w}_s.\tag{4}$$

If the embedding map is locally $L_g$-Lipschitz-smooth, $\|J_w - J_l\| \leq L_g \|h_w - h_l\|$, we have

$$\begin{aligned}\|\nabla_\phi \mathcal{L}\| &= \big| \sigma(d) - 1 \big| \cdot \|\mathbf{w}_s\| \|(J_w - J_l)^\top\| \\ &\leq \big| \sigma(d) - 1 \big| \|\mathbf{w}_s\| \cdot L_g \|h_w - h_l\|.\end{aligned}\tag{5}$$

Secondly, at the component of score layer $\mathbf{w}_s$,

$$\|\nabla_{\mathbf{w}_s} \mathcal{L}\| = \big| \sigma(d) - 1 \big| \cdot \|h_w - h_l\|.\tag{6}$$

Summing both contributions from $\phi, \mathbf{w}_s$, we have the following expression for the norm of Eq.1:

$$\begin{aligned}\|\nabla_\theta \mathcal{L}\| &= \big| \sigma(d) - 1 \big| \, \|\nabla_\theta (r_w - r_l)\| \\ &\leq \big| \sigma(d) - 1 \big| \sqrt{\begin{aligned}&\big( \|\mathbf{w}_s\| L_g \|h_w - h_l\| \big)^2 \\ &+ \|h_w - h_l\|^2\end{aligned}} \\ &= \underbrace{\big| \sigma(d) - 1 \big|}_{\text{prediction error}} \cdot \underbrace{\big( k \|h_w - h_l\| \big)}_{\text{representation distance}},\end{aligned}\tag{7}$$

where $k = \sqrt{1 + \big( L_g \|\mathbf{w}_s\| \big)^2}$.

This decomposition makes clear that the strength of update depends jointly on (1). **prediction error**, i.e., how well the model currently ranks the responses, as captured by the predicted reward difference $d$, and (2). model's sensitivity to the pair, which by derivation in Eq. 7 is governed by **representation distance**.

**Interpretations.** The term $\nabla_\theta(r_w - r_l)$ reflects how $\theta$ is adjusted so that the chosen reward $r_w$ increases while the rejected reward $r_l$ decreases. It captures the *sensitivity of model parameters to this pair*. As shown by decomposing into backbone $\phi$ and score layer $\mathbf{w}_s$ contributions in Eq. 7, this sensitivity is closely related to the representation distance $\|h_w - h_l\|$. Consequently, for two pairs with the same prediction error (i.e., the same $|\sigma(d) - 1|$ term in the BT gradient), the pair with small representation distance $\|h_w - h_l\|$ gives nearly canceling gradients and receives only a weak update, whereas the pair with a large distance receives a stronger update.

To show this concretely, Figure 1 presents two preference pairs with very different representation distances. In the first pair, $(y_w, y_l)_{\text{Safety}}$ are semantically and stylistically dissimilar, leading to a large representation distance term $\|h_w - h_l\|$. In the second pair, $(y_w, y_l)_{\text{Reason}}$ differ by a logical error and are nearly identical in surface form. While humans would prefer the correct $y_w$ over the wrong $y_l$, this pair is harder for the model to separate, producing a smaller representation distance. As a result, Eq. 7 implies that the BT update magnitude for $(y_w, y_l)_{\text{Reason}}$ is naturally weaker than that for $(y_w, y_l)_{\text{Safety}}$, systematically under-training pairs where the model struggles to distinguish. Appendix D presents additional examples drawn from evaluations.

### Quantitative analysis 1: How does representation distance "dominate" update magnitudes?

Figure 2a further isolates the effect of representation distance by controlling for prediction error: we group preference pairs into bins with nearly identical error term $|\sigma(d) - 1|$ (equivalently, similar $d = r_w - r_l$). Within each bin, which holds prediction error term nearly fixed, the observed gradient norms increase systematically with representation distance. This provides direct evidence showing that the distance term dominates the learning signal even when the model is "equally wrong."

### Quantitative analysis 2: How prevalent is this representation distance disparity in preference data?

This effect is not limited to isolated cases but extends across datasets. We illustrate this using RewardBench (Lambert et al., 2024), a benchmark for reward models that provides categorized preference data (Chat, Chat-Hard, Safety, and Reasoning), allowing easy understanding for cross-task com-

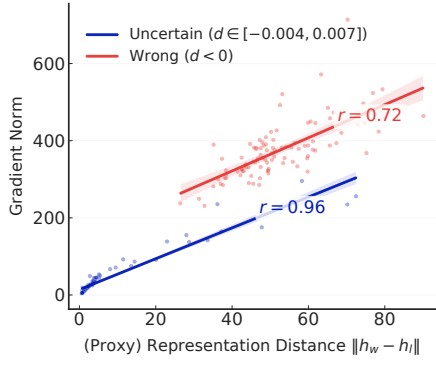
(a) Gradient norms, binned by reward diff. $d$

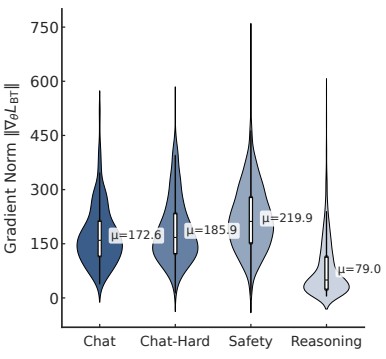
(b) Gradient Norm by Category

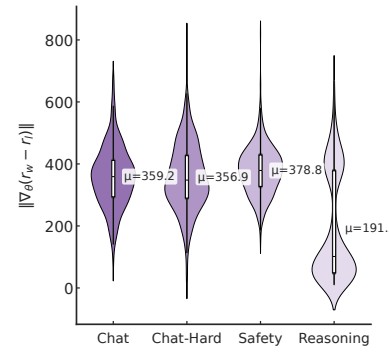
(c) Representation Distance by Category

*Figure 2.* **Gradient Analysis Across Dataset.** (a) Even for pairs with the same prediction error $d$ (i.e., pairs within each color group), its BT gradient norm, hence update magnitude, still scales strongly with representation distance $\|h_w - h_l\|$. (b) BT gradient norms vary substantially across RewardBench categories, with Reasoning receiving the weakest updates. (c) This disparity in (b) aligns with category-wise representation distances: Reasoning pairs are also closest in representation space, while others are farther apart.

parisons. Figure 2b compares the distribution of gradient norms and 2c compares $\|\nabla_\theta(r_w - r_l)\|$ across pairs in RewardBench. The left panel shows that the gradient norms (hence update sizes) under the BT objective vary widely by task. In particular, the *Reasoning category exhibits the smallest average gradient norm*, despite being a domain where fine-grained distinctions are crucial. The right panel reveals why: Reasoning pairs also have the *smallest representation distance*, whereas e.g., Safety and Chat pairs are much farther apart in representation space.

This observation provides dataset-wise evidence for our interpretations. Since BT-loss couples prediction error with representation distance (Eq. 7), updates for small-distance pairs are inherently weaker. Nonetheless, we note that while such small-distance, weak-update cases appear frequently in reasoning tasks under this specific data distribution, the underlying issue is more general: it arises from the representation distance, rather than any single task domain.

**Motivation.** In preference learning, minor differences can decisively determine correctness. For example, a single logical mistake in reasoning tasks can make an otherwise valid solution entirely wrong. In well-curated preference datasets, small representation distance often reflects *meaningful and hard pairs* that the current model fails to separate despite a clear human preference (e.g., correct vs. subtly flawed code). If such pairs inherently receive only weak updates due to small representation distance $\|h_w - h_l\|$, the model under-corrects clear misrankings and receives the least training signal precisely where it struggles most. Conversely, when certain pairs can produce large updates regardless of accuracy, the reward model may overemphasize superficial differences. These observations motivate aligning reward model update strength with *prediction error* rather than representation distance.

### 2.2. Sample-Wise Normalization

To align BT update sizes directly with prediction error, we introduce into the loss function a per-pair weight $w_i(x, y_w, y_l)$ that scales inversely with representation distance. In principle, this weight can be defined as

$$w_i = \mathrm{sg}\left(\frac{1}{\|\nabla_\theta(r_w - r_l)\|}\right), \qquad (8)$$

where $\mathrm{sg}(\cdot)$ denotes the stop-gradient operator. Thus the gradient norm simplifies to

$$\left\|\nabla_\theta\hat{\mathcal{L}}\right\| = w_i \left|\sigma(d) - 1\right| \left\|\nabla_\theta(r_w - r_l)\right\| = \left|\sigma(d) - 1\right|. \quad (9)$$

As $w_i$ directly removes the multiplicative factor from the representation distance, the update size does not depend on this magnitude.

**Representation Distance Proxy.** Computing $w_i$ is impractical at scale, as it requires additional backward operations to obtain per-sample gradient information. Instead, analysis in Sec 2.1 establishes that it is well-approximated by the final-layer representation distance: $\|\nabla_\theta(r_w - r_l)\| \propto \|h_w - h_l\|$ (Eq. 7). We therefore consider $\|h_w - h_l\|$ as a direct proxy, and define the pair-wise weight

$$\tilde{w}_i = \mathrm{sg}\left(\frac{1}{\|h_w - h_l\|}\right), \qquad (10)$$

This proxy offers three key advantages: (1) **Efficient.** $(h_w - h_l)$ is available from the forward pass without additional computations. (2) **Well-founded.** This proxy is not just approximate but *exact* at the linear score head (Eq. 6). Since the score head directly produces the reward prediction, it is natural to view its gradient as a normalization signal. (3) Admits an **intuitive interpretation**. $\|h_w - h_l\|$ corresponds to the final-layer representation distance between the response pair. Small representation gaps often

arise when responses are close in surface form or semantics (e.g., the Reasoning pair in Figure 1), which are precisely the fine-grained distinctions that are hardest for the model to separate. Normalizing by $\frac{1}{\|h_w - h_l\|}$ prevents further suppressing the update strength by upweighting such subtle but decisive pairs. Conversely, highly dissimilar responses that naturally induce large BT updates and are therefore downweighted, reducing the tendency to overemphasize superficial differences in representation space.

The correlation between this proxy and full representation distance $\|\nabla_\theta(r_w - r_l)\|$ is shown in Appendix F.

**Stabilizing Scale Drift.**   Since the scale of embeddings can vary during training, raw values of $\tilde{w}$ may drift and tighten or loosen normalization inadvertently (Appendix G). To make $\tilde{w}$ invariant to such scale drift, we use an exponential moving average (EMA) of the batch statistics.

Under this formulation, we do not eliminate the magnitude of the representation-distance component entirely, but instead rescale each pair relative to the running mean. Let $\mu_t$ denote the tracked EMA estimates of the mean embedding difference at optimization step $t$. We define the normalization weight for a training pair $(y_w, y_l)$ relative to $\mu_t$,

$$\tilde{w}_t(y_w, y_l) = \mu_t/(\|h_w - h_l\| + \epsilon) \quad (11)$$

where $\epsilon$ is a small constant for numerical stability.

Let $\hat{\mu}_t = \mathbb{E}_{\text{batch,t}}[\|h_w - h_l\|]$ denote the mean representation difference in the current batch, the EMA estimate is then updated as

$$\mu_{t+1} \leftarrow \beta\mu_t + (1-\beta)\hat{\mu}_t, \quad (12)$$

with $\beta = 0.9$ by default. Using the EMA-relative scaling $\mu_t$ (rather than a fixed constant) adapts the normalization strength over time while smoothing batch noise, giving weights that are stable and adaptive to distributional shifts. Regardless of the backbone's representation scale drifts, $\tilde{w}$ stays near unity on average and the effective loss scale remains stable.

**Final Objective.**   Concretely, the NORMBT objective incorporates both the proxy for representation distance and EMA-based stabilization. For a preference sample $s = (x, y_c, y_r)$, it is given by

$$\mathcal{L}_{\text{NormBT}}(\theta) = -\mathbb{E}_{s \sim D}\left[\tilde{w}(y_w, y_l) \cdot \log \sigma(r_w - r_l)\right]. \quad (13)$$

In effect, NORMBT rescales the gradient contribution of training pairs such that small-distance pairs are not overshadowed by large-distance pairs.

## 3. Experiments

We conduct comprehensive experiments to evaluate NORMBT against BT baselines, compare downstream performance in Section 3.3, followed by ablation studies in Section 3.5 and analysis of performance gains in Section 3.4.

### 3.1. Setup

**Base Models.** Following standard practice in reward modeling as sequence classifier, we attach a linear score head to a pretrained LLM backbone to output scalar rewards. We experiment with two open-source base models: **gemma-2b-it** (Team et al., 2024) and **Llama-3.2-3B-Instruct**[1]. These choices evaluate NORMBT across distinct backbone families and capacities.

**Datasets.** We train reward models on two pairwise preference datasets, yielding a total of four experimental settings (two backbones × two dataset): (1). **Unified-Feedback**[2]. This is a large and diverse collection of pairwise feedback aggregated from multiple open-source datasets. We use a random subset of 80K preference pairs from Unified-Feedback. (2). **Skywork-Reward-Preference-80K-v0.2** (Liu et al., 2024a). This is a curated dataset of 80K high-quality preference pairs covering diverse capability and knowledge domains. It is the decontaminated version from Skywork-Reward-Preference-80K-v0.1 (Liu et al., 2024a) to remove overlaps with the evaluation benchmark RewardBench, ensuring no contamination between training and evaluation.

**Baselines.** We compare NORMBT against several reward model loss objectives:

1. **BT Baseline:** standard Bradley–Terry loss in Eq. 1.
2. **BT + Margin** (Touvron et al., 2023): introduces the ground-truth reward margin $m$ into the BT loss.
3. **BT + Margin (outside)** (Wang et al., 2025): variant that introduces margin $m$ as a scale factor to the loss.
4. **BT + label smoothing** (Liu et al., 2024a): replaces hard binary targets with soft labels as a regularizer.

The two margin-based variants explicitly modify the prediction-error term by injecting ground-truth reward differences, thereby adjusting update strength according to external signals. Label smoothing also alters the prediction-error term, but by uniformly reducing its magnitude. These methods serve as natural comparisons since they manipulate the error strength, whereas our approach targets representation distance. Detailed formulation and comparison with NORMBT are included in Appendix B.

For all BT baselines, we conduct an extensive grid search over learning rates and report the best-tuned results. Additional training details are included in Appendix B.

---

[1] meta-llama/Llama-3.2-3B-Instruct
[2] llm-blender/Unified-Feedback

*Table 1.* Results on RewardBench from four settings (two base models × two datasets)

*(a)* Models trained from Unified-Feedback (80K)

| Reward model | Chat | Chat-Hard | Safety | Reasoning | Average |
|---|---|---|---|---|---|
| Base Model: gemma-2b-it | | | | | |
| BT (baseline) | 95.25 | 40.35 | 77.97 | 75.41 | 72.25 |
| BT + margin | 95.81 | 37.50 | 78.65 | 72.98 | 71.23 |
| BT + margin out | 96.09 | 38.38 | 77.57 | 78.09 | 72.53 |
| BT + label smooth | 93.85 | 37.72 | 75.95 | 72.28 | 69.95 |
| NORMBT (ours) | 95.81 | 39.80 | 77.97 | 80.71 | **73.57** |
| Base Model: Llama-3.2-3B-Instruct | | | | | |
| BT (baseline) | 95.53 | 49.89 | 81.69 | 71.70 | 75.24 |
| BT + margin | 97.77 | 47.59 | 81.28 | 69.97 | 74.15 |
| BT + margin out | 96.65 | 45.83 | 84.05 | 72.81 | 74.84 |
| BT + label smooth | 95.53 | 49.78 | 79.73 | 72.52 | 74.39 |
| NORMBT (ours) | 96.93 | 49.78 | 84.19 | 76.93 | **76.96** |

*(b)* Models trained from Skywork-Reward-Preference-80K-v0.2

| Reward model | Chat | Chat-Hard | Safety | Reasoning | Average |
|---|---|---|---|---|---|
| Base Model: gemma-2b-it | | | | | |
| BT (baseline) | 81.28 | 73.36 | 82.43 | 77.46 | 78.63 |
| NORMBT (ours) | 83.80 | 73.46 | 82.50 | 80.71 | **80.12** |
| Base Model: Llama-3.2-3B-Instruct | | | | | |
| BT (baseline) | 86.03 | 78.29 | 89.86 | 67.05 | 80.31 |
| NORMBT (ours) | 83.80 | 78.73 | 88.78 | 74.60 | **81.48** |

## 3.2. Main Results

Table 1a reports the performance of all reward models trained on Unified-Feedback (80K) and the two base models. And Table 1b presents the results trained on Skywork-Reward-Preference-80K-v0.2. Across all four settings, the NORMBT objective consistently outperforms the BT baseline, showing that mitigating representation-distance bias improves reward modeling.

**NORMBT vs. BT.** In particular, strong performance gains are observed in the Reasoning category, where NORMBT improves accuracy by more than $5\%$ on average. This is consistent with our analysis: Reasoning pairs tend to have smaller representation distances compared to other categories (Figure 2b). Under BT, the small-distance region receives inherently weak updates, even when misranked. NORMBT explicitly alleviates this by upweighting learning signals for small-distance regions and reducing over-emphasis on large-distance regions. By removing this representation-driven effect in parameter updates, NORMBT provides stronger updates that are more aligned with prediction errors. As a result, the largest gains naturally appear in categories dominated by fine-grained distinctions, which are otherwise suppressed.

**NORMBT vs. BT-variants.** The margin-based baselines (BT+Margin, BT+Margin out) do not yield consistent improvements over the vanilla BT loss. Although these meth-

ods introduce ground-truth margins to directly supervise the strength of the prediction-error term, they do not correct for the scaling bias introduced by representation distance. Cases where margin-based training hurts performance suggest possible overfitting to noisy margin annotations. This highlights that simply enriching supervision with ground-truth requires high-quality annotations, and is still insufficient to address the structural bias of BT updates. In contrast, NORMBT requires no external signals beyond the standard pairwise preference data, and outperforms both variants of margin-based baselines. This shows that addressing the representation bias directly is more reliable and broadly applicable.

Lastly, label-smoothing does not improve over the vanilla BT objective. As shown in Eq 18, label-smoothing softens the target distribution by uniformly reducing the prediction-error magnitude in the gradient. While this can help calibration, it also weakens the update strength globally, including pairs with small-representation distance where the gradient is already small. The results further align with this intuition, as the largest drop in performance is reflected in the Reasoning category (e.g., from 75.41 to 72.28 in gemma-2b-it with Unified-Feedback).

## 3.3. Downstream Performance

To evaluate the practical utility of reward models, we assess their performance on Best-of-N (BoN) response selection. Unlike standard benchmarks that measure binary accuracy, BoN directly reflects how an RM guides a policy model among diverse candidate responses, closely mirroring usage in downstream RLHF pipelines.

We evaluate the reward models from Section 3.1 trained on gemma-2b-it and Unified-Feedback (80K). We further applied NORMBT on top of each BT baselines to examine the effectiveness of our method on downstream performance.

Following prior works (Gao et al., 2022; Coste et al., 2023; Yang et al., 2024), we conduct BoN sampling on a 1K held-out test set. For each prompt, we sample $n$ candidate responses, score them using the trained RM and select the highest-scoring response. We then use a gold reward model to evaluate the selected responses, and report the gold score averaged over the 1K prompts. This reflects the true quality of the responses selected by RM. We set the number of responses $n$ ranging from 1 to 402 for each prompt. This roughly responds to the KL-divergence of 0 to 5 from policy model, according to the equation $\text{KL}_{\text{BoN}} = \ln n - \frac{n-1}{n}$. The gold score model[3] is a 7B RM finetuned on the entire Unified-Feedback dataset, following (Yang et al., 2024).

The results are reported in Figure 3. NORMBT consistently achieves higher gold scores, outperforming all BT baselines. Moreover, applying NORMBT on top of each BT variant

---

[3]reward-model-Mistral-7B-instruct-Unified-Feedback

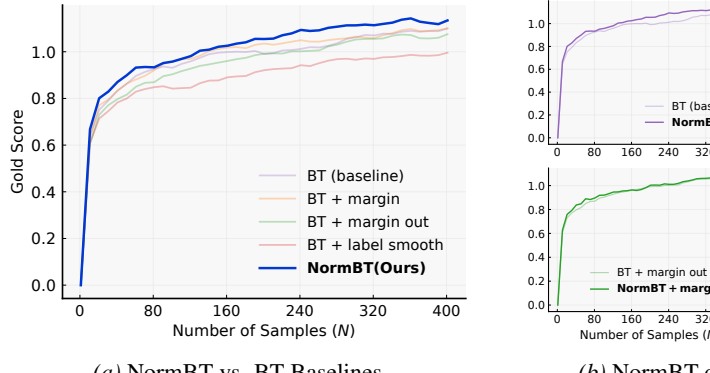
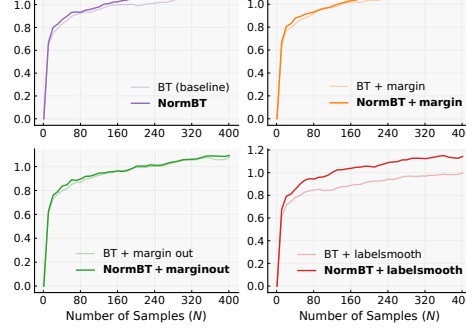

*(a)* NormBT vs. BT Baselines                    *(b)* NormBT on each baseline variant

*Figure 3.* **Best-of-N Selection.** Higher gold score indicates higher quality of the selected responses. (a) NORMBT consistently outperforms all BT baselines, as well as (b) all BT-based counterparts when applied on top of each BT variant.

yields further improvements , showing that balancing the effect of representation distance on learning signals is beneficial to standalone BT objective. The performance gains is most pronouced for label smoothing: smoothing uniformly damps gradients and can further weaken already-small updates for small-distance pairs, whereas NORMBT compensates by upweighting such pairs, leading to a clear boost in BoN gold score. These results demonstrate that NORMBT complements and strengthens existing approaches in ways that carry over to downstream RLHF applications.

### 3.4. Analysis

Beyond the evaluation results presented, we analyze *where* the performance gains of NORMBT originate. Figure 4 shows the subset of RewardBench pairs where the BT Baseline and NORMBT predictions disagree. That is, cases where one model is correct while the other is wrong. The x-axis bins response pairs by their representation distance $\|h_w - h_l\|$, where $h$ are derived from the base model gemma-2b-it, and the y-axis shows the count of these exclusive correct predictions. For ease of interpretation, we divide the range into small, medium, and large distances roughly based on EMA statistics of the embedding distribution; these divisions are intended only for discussion.

The clearest gains of NORMBT over BT appear in the **small-distance regime**. For pairs in this range, BT updates are weak and leave misrankings or ambiguous predictions insufficiently corrected. NORMBT resolves this by normalizing the representation distance factor to an average baseline, ensuring prediction error drives the update. Accordingly, the histogram shows a clear concentration of additional correct predictions from NORMBT in this region, in line with our intuition. Appendix D provides examples from this region that NORMBT predicts correctly while BT doesn't.

In the **medium- and large-distance regimes**, NORMBT still yields general improvements. This is consistent with

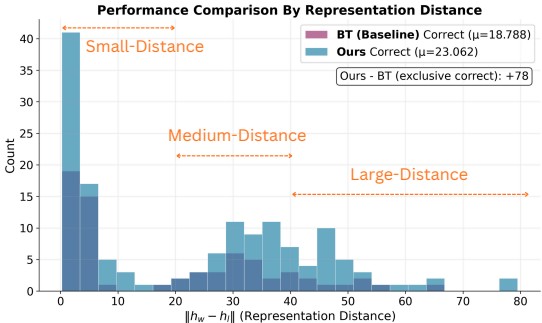

*(a)* Unified-Feedback (80K)

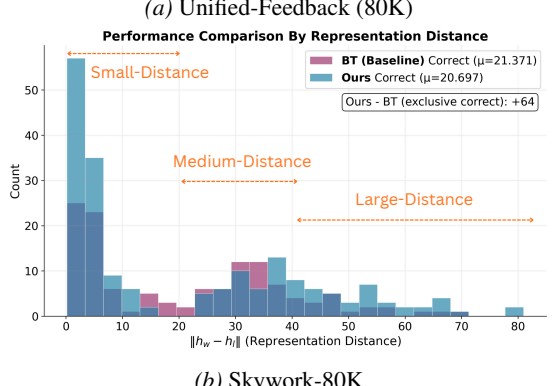

*(b)* Skywork-80K

*Figure 4.* **Analysis.** Comparison of RewardBench pairs where two models disagree, binned by representation distance $\|h_w - h_l\|$. The largest gains appear in the small-distance regime.

our analysis: when representation distance is large, BT already produces strong updates, so the distance term is less of a bottleneck. We do observe small degradations relative to the best baseline, for example, $78.65 \rightarrow 77.97$ on Safety for gemma-2b-it trained on Unified-Feedback (Table 1), suggesting that down-weighting can slightly affect large-distance regions. Overall, NORMBT maintains parity or modest gains, suggesting that improvements on small-distance pairs come at little expense elsewhere.

## 3.5. Additional Results

In this section, we include further empirical studies for our method. We first compare NORMBT with existing training techniques (i.e., Layer normalization, Gradient clipping, and BSR regularization). We then ablate the key design components of NORMBT. Lastly, we apply our method on the BT variants discussed in Section 3 to show that it is complementary with existing works, and report evaluation results on additional benchmarks.

**LayerNorm & Gradient Clipping.** A natural question is whether uniformly normalizing the representation can mitigate distance-induced imbalance, rather than reweighting the gradient contributions. We therefore apply a Layer-Norm (Ba et al., 2016) before the final reward layer. While this yields mild gains in some categories, it underperforms on fine-grained tasks. The key limitation is that LayerNorm stabilizes embedding scale but does not modify the disparity in representation distances, leaving BT gradient dynamics unchanged. Similarly, global gradient clipping (Pascanu et al., 2013) is commonly used to avoid extreme gradients. While this improves optimization stability, the comparison results show that it does not address the relative weighting across pairs that NORMBT targets.

**Batch-wise Sum-to-zero Regularization (BSR).** The work by Hong et al. (2025) is closely related to our analysis, focusing on BT training dynamics under a linear reward head. The key difference lies in the intervention level. BSR connects reward-scale growth with the representation distance between chosen and rejected responses, and controls this effect through a batch-level regularizer on the absolute reward scale. In contrast, our method directly normalizes the distance-dependent component of each pairwise BT update. Thus, BSR suppresses extreme rewards globally, while our method explicitly reweights training across preference pairs. Empirically, NORMBT shows notable gains in the fine-grained Reasoning category (74.40 vs. 80.71), suggesting that adjusting pairwise update allocation is effective.

**Ablations.** We first ablate on similarity measures used for scaling, replacing the last-token embedding L2-distance with (1) cosine similarity and (2) the L2-distance of average-pooled embeddings. Both alternatives underperform compared to NORMBT. This aligns with our theoretical motivation: the last-token embedding is the direct input to the reward head, making its L2 difference an effective proxy for the gradient of the BT score (Eq. 6). Alternative similarity metrics do not reflect this gradient structure. We also remove EMA and observe performance degradation, as the scale of representation distance drift during training (Appendix G). Furthermore, Table 3 reports performance on varying EMA coefficient $\beta$ beyond the default choice of

$\beta = 0.9$ used in the main experiments. While further tuning this parameter may slightly improve the performance, the default or a standard range between $\beta \in [0.8, 0.9]$ provides a robust option overall.

We then apply our method to all BT-based variants for the main experiments discussed in Section 3. The result shows that NORMBT offers consistent improvements across all counterparts, suggesting that distance normalization is complementary to existing BT methods. To further test the generalization beyond RewardBench in Table 1a, we also evaluate one pair of trained counterparts on three additional benchmarks: (1) RM-Bench (Liu et al., 2024d), (2) RewardBench-2 (Malik et al., 2025), and (3) Preference Proxy Evaluations (PPE) (Frick et al., 2024). These benchmarks are generally more challenging than RewardBench (Lambert et al., 2024). In particular, PPE exhibits better correlation with downstream outcomes in post-RLHF LLM performance, and RM-Bench targets fine-grained distinctions in preference pairs. We report the results in Table 4, showing that NORMBT-based method outperforms its BT counterpart. This is consistent with observations from RewardBench and Best-of-N experiments, where our method better distinguishes subtle differences and leads to improvement that can transfer to downstream performance.

*Table 2.* **Ablation Studies.** NORMBT addresses unique training artifacts and is complmentary with existing variants. All reward models are trained from gemma-2b-it on Unified-Feedback (80K).

| Reward model | Chat | Chat-Hard | Safety | Reasoning | Average |
|---|---|---|---|---|---|
| *Existing Training Techniques* | | | | | |
| BT (baseline) | 95.25 | 40.35 | 77.97 | 75.41 | 72.25 |
| BT + BSR | 95.25 | 40.35 | 78.92 | 74.40 | 72.23 |
| BT + LayerNorm | 94.97 | 41.34 | 79.46 | 73.22 | 72.25 |
| BT + Grad. clip | 95.53 | 39.69 | 77.57 | 77.96 | 72.69 |
| *Key Designs* | | | | | |
| NORMBT (No EMA) | 94.97 | 35.31 | 75.81 | 65.04 | 67.78 |
| NORMBT (Avg. pool) | 96.09 | 38.16 | 73.51 | 70.97 | 69.68 |
| NORMBT (Cos. sim.) | 95.53 | 39.47 | 77.16 | 75.37 | 71.88 |
| NORMBT | 95.81 | 39.80 | 77.97 | 80.71 | **73.57** |
| *Apply NORMBT on all variants* | | | | | |
| BT + margin | 95.81 | 37.50 | 78.65 | 72.98 | 71.23 |
| NORMBT + margin | 95.81 | 38.82 | 77.57 | 77.02 | **72.30** |
| BT + margin out | 96.09 | 38.38 | 77.57 | 78.09 | 72.53 |
| NORMBT + margin out | 96.65 | 38.82 | 77.16 | 80.83 | **73.36** |
| BT + label smooth | 93.85 | 37.72 | 75.95 | 72.28 | 69.95 |
| NORMBT + label smooth | 95.53 | 39.25 | 78.11 | 80.96 | **73.46** |

*Table 3.* **EMA Coefficient.** We varying $\beta$ in the EMA stabilizer beyond the default of $\beta = 0.9$ used in main experiments. All models are trained from gemma-2b-it on Unified-Feedback (80K).

| $\beta$ | Chat | Chat-Hard | Safety | Reasoning | Average |
|---|---|---|---|---|---|
| 0.70 | 95.81 | 37.50 | 78.11 | 78.72 | 72.53 |
| 0.80 | 96.09 | 39.25 | 77.97 | 79.50 | 73.20 |
| 0.85 | 96.09 | 39.25 | 78.24 | 80.90 | **73.62** |
| 0.90 | 95.81 | 39.80 | 77.97 | 80.71 | 73.57 |

*Table 4.* **Further evaluations.** We evaluate on three additional benchmarks that are more challenging compared to RewardBench.

| | | | | *RM-Bench* | | | | |
| --- | --- | --- | --- | --- | --- | --- | --- | --- |
| | **Chat** | **Math** | **Code** | **Safety** | **Easy** | **Normal** | **Hard** | **Overall** |
| BT | 44.53 | 54.04 | 50.19 | 82.72 | 84.59 | 59.53 | 29.49 | 57.87 |
| NormBT | 46.60 | 55.81 | 50.05 | 83.45 | 83.81 | 60.66 | 32.46 | **58.98** |

| | | | *RewardBench-2* | | | | |
| --- | --- | --- | --- | --- | --- | --- |
| | **Fact.** | **Prec. IF** | **Math** | **Safety** | **Focus** | **Ties** | **Avg.** |
| BT | 31.26 | 23.75 | 40.44 | 49.56 | 24.24 | 11.22 | 30.08 |
| NormBT | 32.53 | 31.87 | 43.17 | 54.00 | 28.69 | 14.76 | **34.17** |

| | | | *PPE* | | | | |
| --- | --- | --- | --- | --- | --- | --- |
| | **MMLU** | **Math** | **GPQA** | **IFEval** | **MBPP+** | **Pref.** | **Avg.** |
| BT | 55.44 | 43.70 | 44.10 | 54.79 | 56.07 | 57.76 | 50.82 |
| NormBT | 57.06 | 44.47 | 44.30 | 54.73 | 56.09 | 58.30 | **51.33** |

## 4. Related Works

**Reward Model Paradigms.** Reward modeling is central to reinforcement learning from human feedback (RLHF), aligning LLMs with human preferences (Christiano et al., 2017; Stiennon et al., 2022; Bai et al., 2022; Dong et al., 2024). Reward models can be broadly classified into paradigms. The first is discriminative RM, typically with a linear score head trained via a Bradley-Terry (BT) style objective (Wang et al., 2025; Liu et al., 2024a; Yang et al., 2024). Another line of research focuses on generative RMs, aiming to leverage the model's generation ability to produce rationales, critiques, or verifier signals (Mahan et al., 2024; Zhang et al., 2025; Chen et al., 2025b; Zhu et al., 2025; Chen et al., 2025a). Beyond this dichotomy, several branches enrich the supervision signal, such as providing fine-grained feedback (Wu et al., 2023), incorporating multi-objectives (Wang et al., 2024), and providing awareness of distribution or uncertainty (Dorka, 2024; Sun et al., 2025b). In this work, we focus on reward models in the discriminative, BT-based paradigm. While direct preference alignment methods (Rafailov et al., 2024; Meng et al., 2024; Gupta et al., 2025) bypass explicit reward modeling, they implicitly rely on similar BT-style pairwise signals. Our work provides a principled analysis on gradients to show how pairwise signals translate into parameter updates.

**Improvements on Reward Modeling.** Despite the success of reward models in RLHF, a broad literature targets complementary weaknesses in training and deployment. For instance, (Liu et al., 2024b) incorporates ties into BT models, while (Coste et al., 2023) identifies the issue of overoptimization. (Yang et al., 2024) shows that regularizing hidden states helps generalization against distribution shifts. Data-centric improvements curate higher-quality comparisons to enhance reward modeling (Liu et al., 2024c; Cui et al., 2023;

Liu et al., 2025; Shen et al., 2025). A parallel line of work improves evaluation protocols, aiming to better capture the practical utility of reward models (Liu et al., 2024d; Frick et al., 2024; Malik et al., 2025). Our study is orthogonal to these directions, where we address a structural limitation in BT-style updates without modifying data or model architecture. Hong et al. (2025) also analyzes representation distance in RMs and introduces zero-centered reward regularization. Our work analyzes how each pair directly contributes to model updates and proposes to modify the BT gradients explicitly. Finally, theoretical and diagnostic studies (Razin et al., 2025; Sun et al., 2025a) analyze fundamental drawbacks of reward models. Our study is complementary to these works in exploring the limitations as well as further possibilities of the BT objective.

**Reward Model Training Dynamics & Calibration.** A related line of work studies reward model calibration. They aim to make reward scores more reliable under specific sources of bias or uncertainty, such as length, confidence, diversity, or other spurious factors (Huang et al., 2025; Leng et al., 2025; Tao et al., 2024; Halpern et al., 2025). Our work instead focus on training artifacts that arise under the BT-objective, where update magnitude depends not only on prediction error but also on the pair's current representation distance. In the realm of training dynamics, BSR (Hong et al., 2025) studies a closely-related setting that also connects representation distance to nontrivial BT dynamics. In general, our observations are consistent with BSR's findings that the growth in representation distance can be suboptimal. While BSR proposes a batch-level regularizer on the reward scale to mitigate this growth, NormBT directly normalizes the distance-dependent component of each pairwise BT update. This differs from BSR approach and explicitly rebalances how preference pairs contribute to learning.

## 5. Conclusion

In this study, we analyze the widely used Bradley-Terry loss for reward modeling and identify a key limitation in its update dynamics. We show that update magnitude depends jointly on (1) prediction error, and (2) representation distance between the response pair, with the latter introducing biased learning signals that are misaligned with model performance. To this end, we propose NormBT, as a lightweight modification to BT-loss through pair-wise normalization, which rescales gradient contributions and ensures updates are driven by prediction errors. Experiments across various base models and training datasets show consistent performance gains of NormBT over the standard BT objective, especially for challenging tasks with fine-grained distinctions. Our findings provide insights into extracting faithful and efficient learning signals, thereby facilitating preference modeling and LLM alignment.

# Acknowledgements

This work is partially supported by NSF 2048280, 2325121, 2244760, 2331966 and ONR N00014-23-1- 2300:P00001.

# Impact Statement

This work studies the update dynamics of Bradley-Terry (BT) loss for training reward models and proposes NORMBT as an effective normalization scheme. By improving learning from fine-grained preference pairs, our method may lead to more reliable reward signals in downstream applications such as RLHF, potentially improving helpfulness and reducing failure modes where subtle reasoning errors are under-penalized. NORMBT does not by itself address issues of data quality, annotator bias, or the broader risks associated with deploying high-capability language models. Therefore, we encourage practitioners to pair reward-model improvements with careful dataset curation and bias evaluation when applying the method in real-world systems.

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

# A. Limitations & Future Work

**Method.** While NORMBT yields consistent gains on fine-grained tasks, a slight trade-off remains for preference pairs with larger representation distance. Future work could focus on more intricate scaling or decoupling strategies to help mitigate this. Additionally, Unified-Feedback and Skywork-80K are human-annotated or curated datasets; settings with substantial noise or near-duplicate pairs may require careful handling when applying NORMBT to avoid the amplification of noise.

**Evaluation.** Our comparison focuses on BT-based variants. Techniques involving regularization or auxiliary objectives may alter gradient dynamics beyond Eq. 7; exploring how NORMBT interacts with such methods is an interesting direction. We conduct Best-of-N to mirror downstream RLHF usage; full end-to-end policy evaluations (Schulman et al., 2017; Ahmadian et al., 2024) would offer a more comprehensive assessment of how the gains translate into downstream performance.

**Analysis.** Our analysis centers on gradient magnitudes to quantify each pair's contribution to model updates. Incorporating gradient-direction or correlation analysis would provide complementary insight into the underlying representation space.

# B. Implementation Details

### B.1. Baseline Details.

This section includes the formulation for each baseline in Sec 3, as well as their relevance to our method.

1. **BT Baseline.** The standard Bradley–Terry objective in Eq 1.

2. **BT + Margin.** (Touvron et al., 2023) For the Unified-Feedback dataset, which provides ratings for the responses, we extract a ground-truth margin $m$ (i.e., difference in ratings) and incorporate it into the BT loss. In this formulation, the margin is introduced inside the log-sigmoid:

$$\mathcal{L}_{\text{Margin}}(\theta) = -\mathbb{E}_{(x,y_w,y_l,m)\sim D}\left[\log\,\sigma\left(r_w - r_l - m\right)\right]. \tag{14}$$

   This variant shifts the effective decision boundary for each pair of chosen and rejected responses according to the ground-truth margin. This baseline tests the effect of injecting ground-truth-based error strength alone, without modifying BT's representation coupling.

3. **BT + Margin (outside)**. (Wang et al., 2025) In the second formulation, the margin is applied outside of log-sigmoid as a multiplicative weight on the per-sample loss:

$$\mathcal{L}_{\text{Margin(out)}}(\theta) = -\mathbb{E}_{(x,y_w,y_l,m)\sim D}\left[m\cdot\log\,\sigma\left(r_w - r_l\right)\right]. \tag{15}$$

   This objective emphasizes the role of prediction error by upweighting pairs with larger ground-truth margins. It tests the effect of a reweighting scheme driven by ground-truth labels, rather than representation distance.

4. **BT + Label Smoothing.** (Liu et al., 2024a) A widely used regularization technique in classification models, where soft labels replace hard binary targets:

$$\mathcal{L}_{\text{LS}}(\theta) = -\mathbb{E}_{(x,y_w,y_l)\sim D}\left[(1-\alpha)\log\,\sigma\left(r_w - r_l\right) + \alpha\left(r_l - r_w\right)\right]. \tag{16}$$

   Differentiating w.r.t. parameters, its gradient is given by

$$\nabla_\theta\mathcal{L}_{\text{LS}} = \left(\sigma(d) - (1-\alpha)\right)\cdot\left(\nabla_\theta r_w - \nabla_\theta r_l\right), \tag{17}$$

   where $\alpha = 0$ recovers the BT objective. Note that since $\sigma(d) \in [0,1]$, its norm is

$$\left\|\nabla_\theta\mathcal{L}\right\| = \left|\sigma(d) - (1-\alpha)\right|\left\|\nabla_\theta r_w - \nabla_\theta r_l\right\|$$
$$= \Big(\underbrace{1 - \sigma(d)}_{\text{BT magnitude}} - \alpha\Big)\left\|\nabla_\theta r_w - \nabla_\theta r_l\right\|. \tag{18}$$

   From this perspective, label smoothing adjusts the strength of BT gradient from the prediction error term, by further reducing its magnitude by $\alpha$. This offers another relevant and generic method for our comparison.

In summary, the margin-based variants modify the prediction-error term by incorporating ground-truth reward differences, effectively adjusting update strength using external supervision. Label smoothing also acts on the prediction-error term, but does so by uniformly reducing its magnitude across all pairs. These approaches serve as natural baselines that manipulate error strength, while our method addresses a distinct factor by normalizing representation distance.

## B.2. Training Details.

We implement all methods based on transformers (Wolf et al., 2020) and trl (von Werra et al., 2020), developing from the work of GRM (Yang et al., 2024). To use the Unified-Feedback dataset, we downsample the training data from the "all" set. All reward models are trained with the default reward head, as a linear layer with shape (hidden size, 1). For label smoothing, $\alpha$ is set to 0.1 by default. For all BT baselines, we conduct an extensive grid search over learning rates and report the best-tuned results. For BT-based variants (margin, margin-out, label smoothing), we search over a range of learning rates centered around the best value found for BT and report the best configuration. All models are trained for 1 epoch with full parameter tuning. We truncate the inputs over 4096 tokens, and an effective batch size of 256 with gradient accumulation. Our experiments are conducted using NVIDIA RTX A6000 49G.

*Table 5.* Key hyperparameter details in reward model training.

| Basic information | |
|---|---|
| Datasets | Unified-Feedback (80K), Skywork-Reward-Preference-80K-v0.2 |
| Base models | gemma-2b-it, Llama-3.2-3b-Instruct |
| Quantization for training | bf16 |
| Optimizer | sgd |
| Momentum | 0.9 |
| Batch size | 256 |
| Gradient accumulation | 4 |
| Learning Rate Scheduler | cosine |
| Warmup Ratio | 0.03 |

## C. Experiment Error Analysis

To assess the robustness of our main results reported in Table 1, we conduct an expanded evaluation on five random seeds for each method. Using a 40K subset of the Unified-Feedback dataset, we train both NORMBT and the BT baseline following the same training configuration described in Appendix B. We compare these 10 models to show the performance improvement of NORMBT over the BT baseline. The results reported in Table 6 and Figure 5 are consistent with the main experiments, with higher average scores across RewardBench and particularly strong gains in the Reasoning category.

*Table 6.* **Error Analysis on Ten Models.** Performance on RewardBench over models trained on five random seeds.

| Reward Model | Chat | Chat Hard | Safety | Reasoning | Average |
|---|---|---|---|---|---|
| BT (Baseline) | 94.69 | 37.48 | 75.11 | 65.34 | 68.15 |
| NORMBT (Ours) | 95.03 | 38.00 | 74.95 | 70.50 | 69.62 |
| **Improvement** | +0.34% (±0.19) | +0.52% (±0.40) | -0.16% (±0.49) | +5.16% (±2.13) | +1.46% (±0.61) |

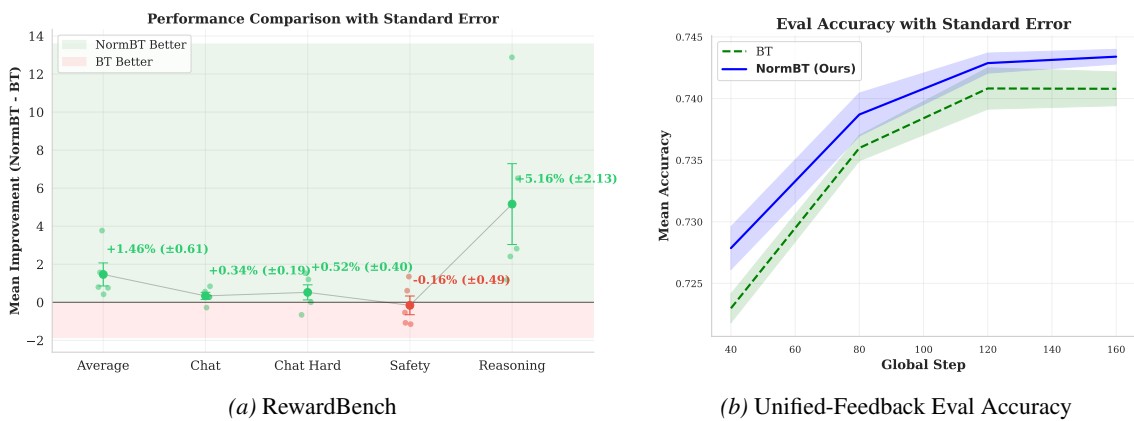

*(a)* RewardBench          *(b)* Unified-Feedback Eval Accuracy

*Figure 5.* **Error Analysis on Ten Models.** Detailed performance of models trained on five random seeds with standard errors on (a) RewardBench categories, and (b) Unified-Feedback Evaluation.

# D. Additional Examples

**Small Representation Distance Pairs.** This section follows up on the performance comparison between BT Baseline and NORMBT in Figure 4. We provide example instances of "Small-Distance" response pairs from RewardBench where NORMBT predicts correctly while BT Baseline does not. This gives a clear illustration to pinpoint the preference pairs where NORMBT gives the largest performance gains.

---

### Pair 1 (Go)

**Prompt.** Write a Go function `func LargestPrimeFactor(n int) int` to solve the following problem: Return the largest prime factor of n. Assume n > 1 and is not a prime.

```
>>> LargestPrimeFactor(13195)
29
>>> LargestPrimeFactor(2048)
2
```

---

```go
isPrime := func(n int) bool {
    for i := 2; i < int(math.Pow(float64(n), 0.5)+1); i++ {
        if n%i == 0 {
            return false
        }
    }
    return true
}

largest := 1
for j := 2; j < n + 1; j++ {
    if n % j == 0 && isPrime(j) {
        if j > largest {
            largest = j
        }
    }
}
return largest
}
```

---

```go
isPrime := func(n int) bool {
    for i := 2; i < int(math.Pow(float64(n), 0.5)+1); i++ {
        if n%i == 0 {
            return false
        }
    }
    return true
}

largest := 1
for j := 2; j < n + 1; j++ {
    if n % j == 0 && isPrime(n) {
        if j > largest {
            largest = j
        }
    }
}
return largest
}
```

**Pair 2 (JavaScript)**

**Prompt.** Write a JavaScript function `const isBored = (S)` to solve the following problem.
You'll be given a string of words, and your task is to count the number of boredoms. A boredom is a sentence that starts with the word `"I"`. Sentences are delimited by `'.'`, `'?'` or `'!'`.

```javascript
let t = 0

if (S[0] == 'I' && S[1] == ' ') { t = 1 }

for (let i = 0; i < S.length; i++) {
    if (S[i] == '.' || S[i] == '!' || S[i] == '?') {
        if (S[i + 1] == ' ' && S[i + 2] == 'I' && S[i + 3] == ' ') {
            t++
        }
    }
}
return t
}
```

```javascript
let t = 0

if (S[0] == ' ' && S[1] == 'I') { t = 1 }

for (let i = 0; i < S.length; i++) {
    if (S[i] == '.' || S[i] == '!' || S[i] == '?') {
        if (S[i + 1] == ' ' && S[i + 2] == 'I' && S[i + 3] == ' ') {
            t++
        }
    }
}
return t
}
```

**Pair 3 (Go)**

**Prompt.** Write a Go function `func NumericalLetterGrade(grades []float64) []string` to solve the following problem.
It is the last week of the semester and the teacher has to give the grades to students. The teacher has been making her own algorithm for grading, but she has lost the code she used for grading. You are given a list of GPAs and must output a list of letter grades using the following table:

```
GPA    | Letter grade
4.0    | A+
...
0.0    | E
```

```go
letter_grade := make([]string, 0, len(grades))
for _, gpa := range grades {
    switch {
    case gpa == 4.0:
        letter_grade = append(letter_grade, "A+")
    case gpa > 3.7:
        letter_grade = append(letter_grade, "A")
    case gpa > 3.3:
        letter_grade = append(letter_grade, "A-")
    case gpa > 3.0:
        letter_grade = append(letter_grade, "B+")
    case gpa > 2.7:
        letter_grade = append(letter_grade, "B")
    case gpa > 2.3:
        letter_grade = append(letter_grade, "B-")
    case gpa > 2.0:
        letter_grade = append(letter_grade, "C+")
    case gpa > 1.7:
        letter_grade = append(letter_grade, "C")
    case gpa > 1.3:
        letter_grade = append(letter_grade, "C-")
    case gpa > 1.0:
        letter_grade = append(letter_grade, "D+")
    case gpa > 0.7:
        letter_grade = append(letter_grade, "D")
    case gpa > 0.0:
        letter_grade = append(letter_grade, "D-")
    default:
        letter_grade = append(letter_grade, "E")
    }
}
return letter_grade
}
```

```go
letter_grade := make([]string, 0, len(grades))
for _, gpa := range grades {
    switch {
    case gpa == 4.0:
        letter_grade = append(letter_grade, "A+")
    case gpa > 3.7:
        letter_grade = append(letter_grade, "A")
    case gpa > 3.3:
        letter_grade = append(letter_grade, "A-")
    case gpa > 3.0:
        letter_grade = append(letter_grade, "B+")
    case gpa > 2.7:
        letter_grade = append(letter_grade, "B")
    case gpa > 2.3:
        letter_grade = append(letter_grade, "B-")
    case gpa > 2.0:
        letter_grade = append(letter_grade, "C+")
    case gpa > 1.7:
        letter_grade = append(letter_grade, "C")
    case gpa > 1.3:
        letter_grade = append(letter_grade, "C-")
    case gpa > 1.0:
        letter_grade = append(letter_grade, "D+")
    case gpa > 0.7:
        letter_grade = append(letter_grade, "D")
    case gpa > 0.0:
        letter_grade = append(letter_grade, "D-")
    default:
        letter_grade = append(letter_grade, "E+")
    }
}
return letter_grade
}
```

# E. Intuition: Large- vs. Small-Distance Pairs

Small representation distance can arise from two reasons: (1) **uninformative or noisy pairs**, such as near-duplicate responses that convey little preference signal; or (2) **difficult but meaningful pairs**, where the backbone fails to separate two responses despite clear human preference signal (e.g., correct vs. subtly flawed code). As illustrated in Figure 2b, such case (2) pairs are common in practice, and Section D provides concrete examples.

This distinction matters because BT loss treats both cases identically: small-distance pairs always receive small gradients. While suppressing case (1) is desirable, suppressing case (2) is harmful, as these are precisely the pairs containing the fine-grained distinctions that the reward model should learn but inherently struggles to separate with the current $\Phi$.

In our datasets (UnifiedFeedback and Skywork-80K), most pairs are human-annotated or carefully curated, and following (Yang et al., 2024), pairs with identical ratings are filtered out. This substantially reduces the prevalence of case (1), making small-distance pairs far more likely to reflect case (2) of the difficult, fine-grained distinctions that RM should learn. Viewed through this lens, NORMBT effectively *upweights hard, meaningful pairs* whose gradients would otherwise be suppressed due to representation limitations in the current model.

# F. Full vs. Proxy Representation Distance

This section analyzes empirically the relationship between the full representation distance $\left\| \nabla_\theta (r_w - r_l) \right\|$ and its proxy $\|h_w - h_l\|$ as defined in Sec 2.1. Figure 6 plots the correlation between these two quantities. For gemma-2b-it, the correlation is strong with $r = 0.928$, suggesting that the embedding difference is a highly reliable stand-in for the full representation distance. For Llama-3.2-3b-Instruct, the correlation is more modest at the beginning of reward model training ($r = 0.682$; shown in Figure 6b) but increases to $r = 0.932$ after some initial training. This may indicate that the proxy becomes more accurate once the pretrained model begins adapting to the reward modeling objective.

These observations support the theoretical analysis in Eq. 7, which establishes the connection between $\left\| \nabla_\theta (r_w - r_l) \right\|$ and $\|h_w - h_l\|$. This suggests the strategy to utilize $\|h_w - h_l\|$ as a proxy is a computationally efficient and empirically validated substitute for the full distance, avoiding costly backward passes.

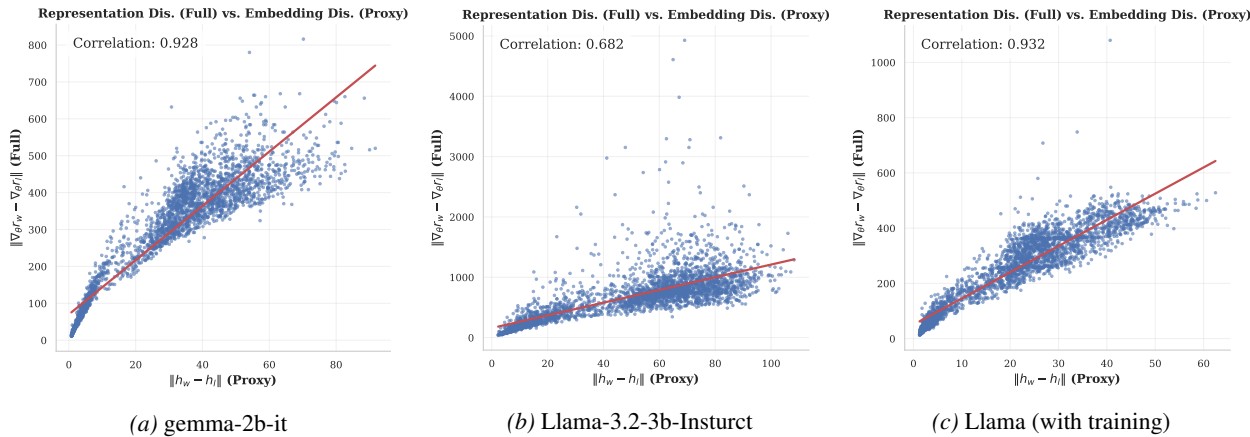

*(a)* gemma-2b-it      *(b)* Llama-3.2-3b-Insturct      *(c)* Llama (with training)

*Figure 6.* **Correlation of Full vs. Proxy Representation Distance.** Both base models show a strong correlation ($r > 0.9$) between the two quantities.

# G. Representation Scale & EMA

One challenge in NORMBT normalization arises from the non-stationary behavior of representation distances $\|h_w - h_l\|$ during training of the reward model. Since the LLM backbone is jointly optimized with the reward head, the learned representation space could evolve considerably.

Figure 7 illustrates this observation: the mean representation distance shifts significantly over the course of optimization, while the variance remains consistently large. And the same behavior is observed across both Unified-Feedback and Skywork-Reward-Preference-80K-v0.2 datasets, as well as for all BT baselines. As a result, although the large variance highlights the potential gradient imbalance, any normalization scheme that relies directly on raw representation magnitudes inherits the instability created by scale shifts.

To mitigate this issue, we incorporate an Exponential Moving Average (EMA) as stated in Eq 12. By tracking a running mean of representation distance, EMA provides a stable and adaptive reference point that smooths out fluctuations in embedding scale while remaining responsive to long-term trends. By normalizing each pair relative to this moving reference, rather than to its raw magnitude, we ensure that weighting reflects relative distinctions among pairs. This yields consistent gradient scaling across training. The benefit of EMA is also reflected in the performance gap observed in ablation studies shown in Table 2.

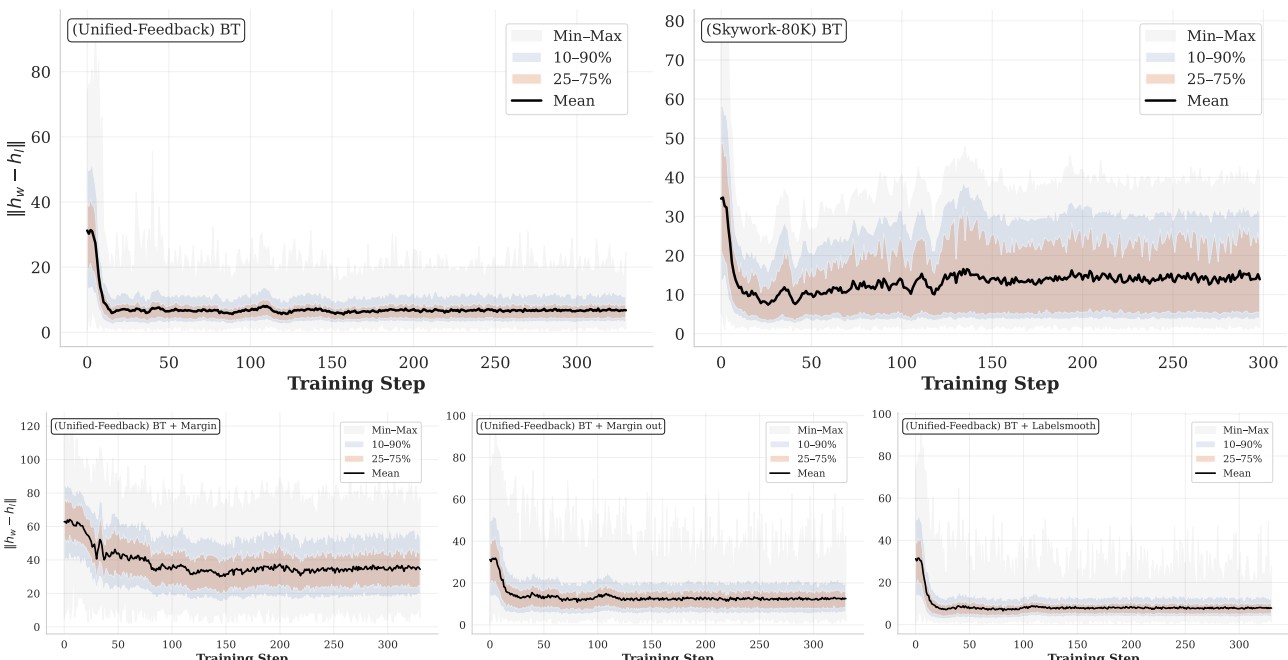

*Figure 7.* **Dynamics of Representation Distance** during RM Training for standard BT on Unified-Feedback and Skywork-80K (top), and all BT variants on Unified-Feedback (bottom). For each method, we report the mean, 25–75 percentile, 10–90 percentile, and full min–max range. Across all settings, the mean distance shifts substantially, while the variance remains high.

