# OpenReview forum: "When Distance Distracts: Representation Distance Bias in BT-Loss for Reward Models"
_ICML.cc/2026/Conference — ICML 2026 regular_

### Official Review · Reviewer_njEP · 2026-02-25

**Soundness:** 3
**Presentation:** 3
**Significance:** 3
**Originality:** 3
**Overall Recommendation:** 4
**Confidence:** 4

**Summary:**

This paper analyzes the gradient dynamics of the Bradley-Terry (BT) loss used in reward model training and identifies that the update magnitude depends not only on prediction error but also on the representation distance between chosen and rejected responses. This coupling leads to a bias: pairs with small representation distance (e.g., reasoning tasks where correct and incorrect solutions differ by a subtle error) receive disproportionately weak updates, while pairs with large representation distance (e.g., safety tasks) receive disproportionately strong updates, regardless of prediction accuracy. To address this, the authors propose NormBT, which normalizes the gradient contribution of each preference pair by the representation distance using an EMA-stabilized per-pair weighting scheme. Experiments on gemma-2b-it and Llama-3.2-3B-Instruct with two preference datasets show consistent improvements, particularly on the Reasoning category of RewardBench.

**Compliance With Llm Reviewing Policy:**

Affirmed.

**Key Questions For Authors:**

1. Does the representation distance bias exhibit different characteristics at larger model scales (e.g., 7B or 13B)? Larger models may produce more separable representations, which could either mitigate or exacerbate the identified bias. Could you provide any analysis or intuition on this?

2. Have you considered applying NormBT to Direct Preference Optimization (DPO) or its variants (SimPO, IPO)? Since DPO uses implicit reward modeling with a similar BT-style loss, the representation distance bias may also manifest there. If you have any preliminary results, they would strengthen the paper's significance.

3. The ablation in Table 2 shows that cosine similarity underperforms L2 distance as a similarity measure. Could you provide more intuition on why direction-based similarity fails here? Given that cosine similarity is commonly used for representation comparison, this result is somewhat surprising and deserves further explanation.

**Limitations:**

The authors acknowledge some performance degradation in the large-distance regime (e.g., Safety category), which is a fair and honest assessment. A more explicit discussion of the limitation to small-scale models and the lack of end-to-end RLHF validation would be appropriate.

**Strengths And Weaknesses:**

### Strengths

1. The gradient analysis of the BT loss is insightful and well-presented. The decomposition into prediction error and representation distance components (Eq. 7) is clean and provides a clear theoretical motivation for the proposed method. The visualization in Figure 2 effectively demonstrates the representation distance bias across RewardBench categories.

2. The proposed NormBT is elegant in its simplicity: it is a lightweight, drop-in modification to the BT loss that requires no architectural changes and incurs negligible computational overhead. The use of the final-layer representation distance as a proxy for the gradient norm is well-justified both theoretically (exact at the linear score head) and empirically.

3. The experimental analysis in Section 3.5 (Figure 4) is particularly convincing. By examining the subset of RewardBench pairs where BT and NormBT disagree, the authors provide direct evidence that NormBT's gains concentrate in the small-distance regime, which is consistent with the theoretical analysis.

### Weaknesses

1. The experimental evaluation is limited to relatively small base models (gemma-2b-it and Llama-3.2-3B-Instruct). The reward modeling landscape has shifted towards larger models (7B+), and it remains unclear whether the representation distance bias persists or changes character at scale. Experiments with at least one 7B+ model would substantially strengthen the claims.

2. The paper only evaluates reward model quality in isolation (RewardBench scores and BoN selection). However, the ultimate goal of reward models is to guide RLHF training. Without end-to-end RLHF experiments showing that a reward model trained with NormBT leads to better-aligned policy models, the practical impact remains unclear. A downstream RLHF experiment, even at small scale, would be valuable.

3. While the theoretical analysis (Section 2.1) is insightful, the Lipschitz smoothness assumption (Eq. 5) provides only an upper bound. The actual coupling between representation distance and gradient norm may be looser in practice, especially for deep transformer backbones where the Jacobian structure is complex. The paper could benefit from a more careful discussion of when this bound is tight versus loose.

4. The Choice of EMA momentum $\beta = 0.9$ is not ablated. Given that scale drift of embeddings during training is the motivation for EMA (and the no-EMA ablation shows significant degradation), understanding the sensitivity to $\beta$ is important.

---

> ### Author Rebuttal · Authors · 2026-03-31
>
> Thank you for the thoughtful and encouraging comments. We sincerely appreciate the reviewer's recognition for our work. We hope the discussion and new results further address your questions, and we are happy to provide any additional information.
>
> >**W1 & Q1**: Does the representation distance bias exhibit different characteristics at larger model scales (e.g., 7B or 13B)?
>
> We report results of BT vs. NormBT on Gemma-7B-Instruct, trained on a smaller 40K random subset of Unified-Feedback. The results are consistent with the paper, where our method outperforms BT with clear gains in Reasoning. This shows that the same representation distance bias persists at larger model scale, similar to our findings on 2B and 3B models.
>
> |Model|Chat|Chat Hard|Safety|Reasoning|Average|
> |-|-:|-:|-:|-:|-:|
> |BT|97.21|45.72|80.95|67.49|72.84|
> |NormBT|96.51|45.94|80.27|72.55|73.82|
>
>
> >**W2**: A downstream RLHF experiment would be valuable.
>
> Due to time constraints, we add results on PPE, a benchmark shown to correlate well with downstream post-RLHF LLM performance. We further validate BoN experiment results using LLM-as-a-judge, to strengthen the credibility of our downstream evaluation. Both results are reported in our response to Reviewer ZCsc (W2&Q2, W3&Q3 respectively). We will also discuss the lack of end-to-end RLHF more explicitly in limitations as suggested.
>
>
>
> >**W3**: The Lipschitz smoothness assumption (Eq. 5) provides only an upper bound. The paper could benefit from a more careful discussion of when this bound is tight versus loose.
>
> We agree that a clearer discussion of the bound is important. Empirically, our proxy is highly correlated with the true gradient quantity ($r= 0.928$, Appendix F), which suggests that the upper bound is sufficiently faithful in practice. Nonetheless, the bound can be looser when the model's local Jacobian varies substantially across layers or representation space. We will include this discussion in the revised draft.
>
>
> >**W4**: The Choice of EMA momentum ($\beta = 0.9$) is not ablated.
>
> We report an ablation over EMA $\beta$ below. While minor fluctuations exist, all NormBT models outperform BT baseline ($72.25$), with consistent gains in Reasoning ($75.41$ vs. $\geq 78.72$). As normalization can become sensitive or stale to short-term shifts in representation distance, $\beta=0.9$ provides a robust default, while modest tuning can yield slightly better results. We will include this study in the revision.
>
> |β|Chat|Chat Hard|Safety|Reasoning|Average|
> |-:|-:|-:|-:|-:|-:|
> |0.90|95.81|39.80|77.97|80.71|73.57|
> |0.85|96.09|39.25 |78.24|80.90|73.62|
> |0.80|96.09|39.25|77.97|79.50|73.20|
> |0.70|95.81|37.50|78.11|78.72|72.53|
>
> >**Q2**: Applying NormBT to Direct Preference Optimization (DPO) or its variants would strengthen the paper's significance.
>
> We outline a sketch of extending our analysis to DPO. **(1) Loss gradient**: for DPO, $\nabla_\theta L_{\rm DPO} = (\sigma(g_\theta)-1)\ \beta\ \nabla_\theta d_\theta(x,y_w,y_l)$, where $d_\theta=\log \pi_\theta(y_w\mid x)-\log \pi_\theta(y_l\mid x)$. This is directly analogous to Eq.(2), that the update magnitude is shaped by both prediction error and sensitivity term $\|\nabla_\theta d_\theta\|$. **(2) Decomposition of sensitivity**: Analogous to how NormBT decomposes into $\nabla_\theta(r_w-r_l)$ backbone and reward-head (Eq.4-6), DPO consists of backbone and LM-head but now over a sequence model. Since $\log \pi_\theta(y|x)=\sum_t \log \pi_\theta(y_t\mid x,y_{<t})$, DPO sensitivity breaks into token-level contributions of the form $\Big(\frac{\partial z_t}{\partial \theta}\Big)^\top (e_{y_t}-p_t)$, where $z_t=\mathrm{logits}$, $p_t=\mathrm{softmax}(z_t)$, and $e_{y_t}$ is one-hot target token.  **(3) Lipschitz**: under local boundedness assumptions on the hidden-states and logits, derive upper bound on $\|\nabla_\theta d_\theta\|$.
>
> In summary, the key difference is that BT RM uses a reward-head on final-layer representation, while DPO involves an autoregressive sequence modeling from policy update. This allows NormBT to derive a clean proxy, whereas DPO is more structured and requires careful decomposition to derive a proxy for the sensitivity term. We believe this perspective may provide useful insight for future works, and we are happy to expand this discussion in our revised draft.
>
> >**Q3**: Why does direction-based cosine similarity underperform?
>
> The motivation of NormBT arises from gradient magnitude. Eq.(7) shows that BT update magnitude is controlled by representation distance, which is inherently a norm-based quantity. In contrast, cosine similarity is direction-based and discards the norm information. It is therefore less aligned with the quantity identified by our analysis as controlling update strength, which likely explains its weaker performance.

---

> > ### Author Rebuttal · Reviewer_njEP · 2026-04-03
> >
> > Thank you for the thorough rebuttal. The authors have addressed all of my main concerns to a satisfactory degree.
> >
> > For W1/Q1, the Gemma-7B-Instruct results directly confirm that the representation distance bias persists at larger model scale, with NormBT achieving a +5.06 gain on Reasoning (67.49→72.55) comparable in character to the 2B/3B findings. This substantially strengthens the scalability claim. For W4, the EMA momentum ablation (β = 0.70–0.90) shows that all settings outperform the BT baseline and that β = 0.85 is a robust default, which is sufficient guidance for practitioners. For Q3, the explanation that cosine similarity discards norm information — which is precisely the quantity identified by the gradient analysis as controlling update strength — is principled and convincing.
> >
> > For W2, the PPE benchmark and LLM-as-a-judge BoN validation serve as reasonable downstream proxies given the difficulty of running full RLHF experiments, and I appreciate the authors' willingness to add an explicit limitations discussion in the revision. For W3, the empirical correlation between the Lipschitz proxy and the true gradient quantity (Appendix F) provides sufficient practical grounding, even if the theoretical bound is not always tight.
> >
> > I maintain my score of 4 (weak accept). The paper's gradient analysis is insightful and the NormBT fix is elegant; the rebuttal addressed the main empirical gaps that concerned me.

---

> > > ### Author Response · Authors · 2026-04-08
> > >
> > > We sincerely thank the reviewer for the careful evaluation throughout the discussion.
> > >
> > > We are grateful for the positive assessment of our work, and we are glad that the concerns have been fully resolved. The comments have also helped us strengthen the results, and present a clearer and more detailed view into our method.

---

### Official Review · Reviewer_zceP · 2026-03-09

**Soundness:** 3
**Presentation:** 3
**Significance:** 2
**Originality:** 2
**Overall Recommendation:** 3
**Confidence:** 3

**Summary:**

This paper identifies a representation distance bias in BT loss for reward model training: the gradient norm scales not only with prediction error but also with the L2 distance between chosen/rejected hidden representations, causing high-distance pairs to dominate updates while low-distance pairs are under-optimized.
The proposed NormBT method reweights each pair by the inverse of its representation distance with EMA stabilization.
Experiments on 2–3B models show improvements on RewardBench.

**Compliance With Llm Reviewing Policy:**

Affirmed.

**Final Justification:**

I appreciate the additional experiments the authors provided, I still think the rational behind the method need further investigation.

**Key Questions For Authors:**

1. The theoretical analysis relies on Lipschitz & linear assumptions and yields only an upper bound on the gradient norm. How does the analysis extend to the nonlinear regime? Or can a tighter bound be established?
2. How does NormBT perform on other benchmarks beyond RewardBench? Results on 7B+ models would also strengthen the empirical case.
3. Hong et al. (2025) also identified the same root cause. Can you directly compare NormBT against BSR?

**Limitations:**

yes

**Strengths And Weaknesses:**

### Strengths
1. The gradient decomposition is clear and the controlled experiment (Figure 2a, r=0.96) directly supports the identified correlation between representation distance and gradient norm.
2. NormBT is lightweight and easy to implement as a drop-in replacement with improved performance on RewardBench.

### Weaknesses

1. **Theoretical Analysis:** The derivation assumes Lipschitz continuity & linear score head, and yields only an upper bound on the gradient norm. Moreover, the analysis focuses on gradient magnitude only, ignoring gradient direction: amplifying the update for ambiguous pairs whose gradient direction is unreliable may not help.

2. **Motivation:** Representation structure is part of what the RM learns. Directly normalizing the representation distance, which discards a critical part of information from the representation space, seems to be counter-intuitive.

3. **Experiments:** Only 2–3B models, two 80K datasets, and a single benchmark (RewardBench). The RM community now has multiple established benchmarks (RM-Bench, RMB, etc.) and routinely evaluates at 7B+ models.

4. **Related work:** Hong et al. (2025, arXiv:2505.07271). That paper derives the *identical* core equation linking BT gradient norm to representation distance (their Eq. 21) and proposes BSR as a solution. The present paper should compare against BSR experimentally.

---

> ### Author Rebuttal · Authors · 2026-03-31
>
> Thank you for the insightful comments and suggestions to help improve our work. We hope the discussion and new results below address your concerns, and we are happy to provide any additional information.
>
> >**W1 & Q1**: The derivation assumes Lipschitz continuity & linear score head, and yields only an upper bound. How does the analysis extend to nonlinear regime?
>
> We agree that deriving a tighter bound is valuable, as the bound can be looser when the model's local Jacobian varies substantially across layers or representation space. Empirically, our proxy is highly correlated with the true gradient quantity (r= 0.928, Appendix F), which suggests that the upper bound is sufficiently faithful in practice. However, we treat this as a valuable future work, and will include it in the revision.
>
> In the case of a nonlinear score head $r(h)=f_\psi(h)$, a first-order expansion gives $r(h_w)-r(h_l)\approx J_f(\bar h)(h_w-h_l)$, where $J_f(\bar h)$ is the Jacobian at $\bar h$. Consequently, $\|r(h_w)-r(h_l)\|\leq \|J_f(\bar h)\| \|h_w-h_l\|$. The BT gradient remains controlled by the representation distance term, up to a multiplicative factor given by the local Jacobian. Therefore, the same gradient behavior is expected to persist, since the effect arises from the pairwise structure of the BT loss itself rather than the linearity assumption. We use a linear head in our analysis because it is standard in reward modeling and yields a clean interpretation.
>
> >**W2**: Representation structure is part of what the RM learns.
>
> We do not claim that representation distance is uninformative. Our concern is that under BT, the *current* learned representation distance acts as an implicit multiplier on update magnitude (Eq.7). This is not always desirable, because a small distance may reflect distinctions that the current model has not yet learned to separate. Therefore, allowing the current representation structure to dictate training strength leads to a self-reinforcing bias: already-separated pairs receive large updates, while fine-grained pairs, precisely where the RM needs stronger signals, receive weak updates.
>
> NormBT uses this distance term to rebalance how pairs contribute to learning, such that training is not overly dominated by pairs that are already easy to separate. In this sense, we use representation distance as training dynamic signals rather than removing it. More broadly, the RM is free to learn the representation structure most useful for the task, since NormBT only rescales updates but does not constrain the representation space itself.
>
>
> >**W3 & Q2**: How does NormBT perform on other benchmarks? Results on 7B+ models would also strengthen the empirical case.
>
> We added 3 new benchmarks (RM-Bench, RewardBench-2, and PPE) in response to Reviewer ZCsc (W2&Q2), and kindly refer the reviewer there due to space limits. In particular, RM-Bench targets fine-grained distinctions and PPE is well-correlated with downstream post-RLHF performance. We further report comparison on 7B model in our response to Reviewer njEP (W1&Q1). NormBT improves over BT-based models on these new evaluations, consistent with findings reported in paper.
>
>
>
> >**W4 & Q3**: Hong et al. (2025) derives the identical core equation. Can you compare NormBT against BSR?
>
> The two works align in the initial setup: both analyze BT with a linear head, and study whether BT induces nontrivial training dynamics. The difference begins in the analysis target and the role of $\|h_w-h_l\|$. We analyze the *full BT gradient*, and derive a relation between the overall update strength and $\|h_w-h_l\|$ (Eq. 7). This motivates an intervention at the level of per-pair updates. In contrast, BSR analyzes only the gradient *with respect to reward head* $W_p$ to connect reward-scale growth with $\|h_w-h_l\|$ (Eq. 19-23), and intervenes with a batch-level regularizer on absolute reward scale (Eq. 25). Thus, BSR softly suppresses extreme rewards, while NormBT directly reweights training across pairs.
>
> Our findings support BSR's claim that growth in $\|h_w-h_l\|$ is problematic (end of Sec 4.1). From the BSR perspective, NormBT can be viewed as addressing the same issue by adjusting pairwise updates: large-distance pairs which naturally receive larger BT updates are downweighted, thereby reducing the tendency for reward-scale growth that BSR controls via regularization.
>
> We report the comparisons below. NormBT shows notable gains in the fine-grained Reasoning category, suggesting that adjusting pairwise update allocation is effective. We thank the reviewer for highlighting this relevant work; it reinforces our findings, and we will include this discussion in the revision.
>
> |Model|Chat|Chat Hard|Safety|Reasoning|Average|
> |-|-:|-:|-:|-:|-:|
> |BSR|95.25|40.35|78.92|74.40|72.23|
> |NormBT|95.81|39.80|77.97|80.71|73.57|

---

> > ### Author Rebuttal · Reviewer_zceP · 2026-04-03
> >
> > Thank you for the detailed clarifications and additional analyses. While the rebuttal addresses some of my concerns, I believe the current evidence is still not sufficient to change my overall assessment, so I am keeping my original score unchanged.

---

> > > ### Author Response · Authors · 2026-04-08
> > >
> > > We sincerely thank the reviewer for the helpful feedback in evaluating our work.
> > >
> > > We are glad that the concerns have been fully resolved. We appreciate the opportunity to elaborate on the motivation of our work. This discussion helped strengthen the results and improve the positioning of our method.

---

### Official Review · Reviewer_ZCsc · 2026-03-11

**Soundness:** 2
**Presentation:** 3
**Significance:** 2
**Originality:** 2
**Overall Recommendation:** 3
**Confidence:** 4

**Summary:**

The paper analyzes the gradient dynamics of the Bradley–Terry loss used in reward model training and shows that gradient magnitudes are influenced by representation distance between response embeddings, which can bias learning toward distant pairs. To address this issue, the authors propose NormBT, a simple normalization method that rescales pairwise updates. Experiments across several LLM backbones demonstrate consistent improvements, especially on reasoning subsets of RewardBench.

**Compliance With Llm Reviewing Policy:**

Affirmed.

**Final Justification:**

The authors have adequately addressed my concerns, and the additional experiments on RM-Bench and newer benchmarks are appreciated. However, the overall novelty and significance are somewhat limited. I maintain my original score.

**Key Questions For Authors:**

1. Could the authors explicitly clarify how their problem formulation and the proposed NormBT method differ from, or build upon, existing literature that addresses reward model miscalibration, e.g., margin-based losses, point-wise ranking, or calibration.
2. How does NormBT perform on newer, more challenging evaluation? I most care about RM-bench, because it focus on whether the reward model could tell subtle differences.
3. Would there be some bias with a 7B reward model as the "gold" standard in BoN evaluation? Have the authors considered evaluating the BoN selected responses using a significantly stronger oracle model or LLM-as-a-judge?

**Limitations:**

Yes

**Strengths And Weaknesses:**

- Strength
  - The paper provides a detailed analysis of the BT loss, identifying how representation distance influences update magnitudes.
  - NormBT is lightweight and easy to integrate into existing reward model training pipelines. It does not require architectural changes and introduces minimal computational overhead.
-  Weakness
   - The issue of gradients being affected by representation distance, mentioned in this paper, is seems to be a miscalibration issue that has been discussed  in the training of reward models in RLHF, and has been studied in previous literature. The authors did not explain these works or compare them in their work.
   - RewardBench is an out-of-date benchmark and is not much discriminative as a reward model benchmark. It is hoped that the authors can compare with some newer benchmarks, such as: Rewardbench 2 (https://arxiv.org/abs/2506.01937), RM-bench (https://arxiv.org/abs/2410.16184), and RMB (https://arxiv.org/abs/2410.09893).
   - Even though the BoN could review the downstream models' ability in test time; real RLHF training would better illustrate the effectiveness of the reward model. Furthermore, the authors used a 7B reward model as the golden benchmark, which is prone to inaccuracies.

---

> ### Author Rebuttal · Authors · 2026-03-31
>
> Thank you for the insightful comments and suggestions to help improve our work. We hope the discussion and new results below address your concerns, and we are happy to provide any additional information.
>
> >**W1 & Q1**: How does the problem formulation and NormBT differ from, or build upon, existing works on miscalibration (e.g., margin-based losses, point-wise ranking, or calibration)?
>
> NormBT identifies and corrects a structural bias in the BT gradient that is distinct from standard miscalibration or ranking objectives.
>
> (a) **Margin-based losses** [1] introduces a target reward margin between (chosen, rejected). This focuses on the prediction-error term, injecting the ground-truth margin to shift the decision boundary or rescale loss. However, our gradient norm decomposition (Eq. 7) reveals that the update magnitude is a product of the prediction error and the representation distance. Thus margin-based methods fail to correct the structural scaling bias in BT due to representation distance.
>
> (b) **Point-wise ranking** [2] trains RM to produce an absolute score for a response, avoiding the pairwise gradient coupling issue by design. However, pure point-wise training is less common in LLM alignment because human absolute ratings are significantly noisier than relative preferences. We focus on the standard BT framework, and build upon this by analyzing the pairwise training dynamics and adjusting updates across pairs.
>
> (c) **Calibration** [3,4] typically addresses output reward bias, e.g., confidence, style, or other spurious factors. Our work targets bias that arises during BT training, where update magnitude depends not only on prediction error but also on the pair’s current representation distance.
>
> In summary, our work is orthogonal to these existing directions, and we address the BT pairwise structure without modifying data, model architecture, or prediction targets. We would be happy to further elaborate if the reviewer has particular works in mind, or if the intended interpretation differs.
>
> >**W2 & Q2**: Performance on newer, more challenging evaluation, especially on RM-Bench?
>
> We added results for a BT variant on 3 new benchmarks: RM-Bench, RewardBench-2, and PPE [5], showing improvements over BT-based model on all metrics. RM-Bench also highlights NormBT on distinguishing subtle differences, consistent with findings on RewardBench reported in paper.
>
> **RM-Bench**
> | Model   | Chat  | Math  | Code  | Safety | Easy  | Normal | Hard  | Overall |
> |---------|------:|------:|------:|-------:|------:|-------:|------:|--------:|
> | BT      | 44.53 | 54.04 | 50.19 |  82.72 | 84.59 |  59.53 | 29.49 |   57.87 |
> | NormBT  | 46.60 | 55.81 | 50.05 |  83.45 | 83.81 |  60.66 | 32.46 |   58.98 |
>
>
> **RewardBench-2**
> | Model   | Factuality | Precise IF | Math  | Safety | Focus | Ties  | Average |
> |---------|-----------:|-----------:|------:|-------:|------:|------:|--------:|
> | BT      |      31.26 |      23.75 | 40.44 |  49.56 | 24.24 | 11.22 |   30.08 |
> | NormBT  |      32.53 |      31.87 | 43.17 |  54.00 | 28.69 | 14.76 |   34.17 |
>
>
> **PPE**
> | Model   | MMLU Pro | Math  | GPQA  | IFEval | MBPP-Plus | PPE Pref | Average |
> |---------|---------:|------:|------:|-------:|----------:|---------:|--------:|
> | BT      |    55.44 | 43.70 | 44.10 |  54.79 |     56.07 |    57.76 |   50.82 |
> | NormBT  |    57.06 | 44.47 | 44.30 |  54.73 |     56.09 |    58.30 |   51.33 |
>
>
> >**W3 & Q3**: BoN reviews downstream ability, but RLHF training would better illustrate effectiveness. And consider evaluating BoN with a stronger oracle model or LLM-as-a-judge.
>
> We provide LLM-as-a-judge evaluation for all 4 model pairs in BoN. We use GPT-4o-mini as the judge, follow the evaluation prompt in [6], and randomize response order to mitigate position bias. We report the win rates below, where each row compares a NormBT model against its BT-based counterpart. While we do not have RLHF experiments due to time constraint, we added results on PPE above (W2&Q2), which is a benchmark shown to correlate well with downstream outcomes in post-RLHF LLM performance [5].
>
> | Comparison           | Win rate (Ours) | Win rate (BT) |
> |----------------------|----------------:|--------------:|
> | NormBT vs. BT        |           46.02 |         44.82 |
> | Margin               |           51.55 |         38.82 |
> | Margin Out           |           49.85 |         41.00 |
> | Label Smoothing      |           48.19 |         41.82 |
>
> *References*
>
> [1] Llama 2: Open Foundation and Fine-Tuned Chat Models (2307.09288)\
> [2] HelpSteer2-Preference: Complementing Ratings with Preferences (2410.01257)\
> [3] Taming Overconfidence in LLMs: Reward Calibration in RLHF (2410.09724)\
> [4] Post-hoc Reward Calibration: A Case Study on Length Bias (2409.17407)\
> [5] How to evaluate reward models for rlhf (2410.14872)\
> [6] Judging LLM-as-a-Judge with MT-Bench and Chatbot Arena (2306.05685)

---

> > ### Author Rebuttal · Reviewer_ZCsc · 2026-04-03
> >
> > I appreciate the author's response and clarifications. I hope the author will incorporate the clarifications regarding the calibration of other reward models into the main text. However, in the additional experiments provided by the author regarding downstream task performance, the win rate of NormBT relative to BT does not appear particularly encouraging. Overall, considering the overall contribution and novelty of the work, I have decided to maintain my original score.

---

> > > ### Author Response · Authors · 2026-04-03
> > >
> > > Thank you for your valuable follow-up and for your time in evaluating our work. We believe there may have been a misunderstanding due to our table presentation, and we will revise it to make the comparison explicit.
> > >
> > > ---
> > >
> > > ### **1. Clarifying BoN Results**
> > >
> > > In the original table (W3&Q3), each row is a matched-pair comparison between a NormBT model and its corresponding BT-based counterpart under the same training configuration (e.g., baseline, margin, margin out, and label smoothing). In the comparisons, **NormBT consistently outperforms BT in all four settings**, especially with NormBT+margin loss (51.55 vs. 38.82). We present the clarified table below:
> > >
> > > | Comparison | Win rate (NormBT) | Win rate (BT) |
> > > |---|---:|---:|
> > > | NormBT vs. BT | **46.02** | 44.82 |
> > > | NormBT (Margin) vs. BT (Margin) | **51.55** | 38.82 |
> > > | NormBT (Margin Out) vs. BT (Margin Out) | **49.85** | 41.00 |
> > > | NormBT (Label Smooth) vs. BT (Label Smooth) | **48.19** | 41.82 |
> > >
> > > NormBT yields clear and consistent improvements over BT-based baselines across all settings. We will also revise the presentation in paper to make the improvements explicit and avoid ambiguity.
> > >
> > > ### **2. Related Works**
> > >
> > > We also appreciate the reviewer’s suggestion regarding calibration-related methods. To the best of our knowledge, NormBT is the first work to identify the representation-dependent bias in BT gradient, and propose a solution that is orthogonal to existing calibration techniques. We are happy to include a more detailed discussion in the main text as suggested, along with concrete examples to clarify the differences, complementarity, and significance of our approach.
> > >
> > > ---
> > >
> > > We thank the reviewer again for all the thoughtful feedback. If this clarification resolves the concern, we would greatly appreciate any further consideration of our work.

---

### Decision · Program_Chairs · 2026-04-30

**Decision:**

Accept (regular)

**Comment:**

### Reviews and discussion

Scores: 3, 3, 4

Reviewer njEP (4, conf 4):
- Found the gradient analysis "insightful," NormBT "elegant in its simplicity," and the disagree-pair analysis (Figure 4) "particularly convincing." Concerns about small model scale and lack of RLHF results were addressed: 7B results confirmed the bias persists, PPE provides a downstream proxy, EMA ablation shows robustness. Concluded the rebuttal was "principled and convincing" and the additions "substantially strengthen" the work.

Reviewer ZCsc (3, conf 4):
- Concerned about positioning (is this a known miscalibration issue?) and outdated benchmarks. Marked "Fully resolved" after the authors clarified that NormBT is distinct from calibration and margin-based methods and provided RM-Bench, RewardBench-2, and PPE results. Still maintained score because "the overall novelty and significance are somewhat limited." Filed Final Justification.

Reviewer zceP (3, conf 3):
- Raised the most specific technical concerns: the Lipschitz bound is only an upper bound (not tight), normalizing by representation distance may be counter-intuitive in some cases, narrow experimental scope, and concurrent work BSR derives the "identical core equation." Marked "Fully resolved" after the authors provided a BSR comparison (NormBT outperforms on Reasoning: 80.71 vs 74.40), nonlinear extension discussion, and additional benchmarks. Maintained score: "the current evidence is still not sufficient to change my overall assessment."

### Assessment

I recommend weak accept.

The gradient decomposition (Eq. 7) identifying how representation distance biases BT update magnitude is precise, with r=0.928 correlation between the proxy and true gradient norm. NormBT is practical and easy to adopt. The rebuttal covered real ground: 3 new benchmarks, 7B scaling, BSR comparison, EMA ablation, LLM-as-judge evaluation. Reviewer njEP engaged deeply and supports acceptance.

The concern I share with the two score-3 reviewers is about novelty and significance. The concurrent BSR paper (Hong et al., 2025) derives the same equation linking BT gradient norm to representation distance. While NormBT and BSR propose different fixes (per-pair normalization vs. batch-level regularization), sharing the diagnostic insight limits this paper's originality claim. The empirical improvements are consistent but modest, typically 1-2% on average across benchmarks, with the largest gains concentrated in Reasoning categories. Without end-to-end RLHF validation, the practical impact remains uncertain.

On the four review dimensions: soundness is the paper's strong suit (correct analysis, validated proxy, clean ablations). Presentation was rated positively by all three reviewers. Significance is the weak point, given the modest effect sizes and missing downstream RLHF evidence. Originality is undercut by concurrent work.

All three reviewers marked "Fully resolved" but maintained their original scores. This pattern signals that the concerns are about the contribution's weight rather than technical correctness. The paper would benefit from clearer positioning relative to BSR and stronger downstream validation before publication.